# WHY DO LLMS FAIL AT ARITHMETIC LOGIC? A DIAGNOSIS OF LIMITS ON SEQUENTIAL COMPUTATION

## ABSTRACT

Despite their power as general sequence processors, Transformers systematically fail at **simple sequential arithmetic tasks like counting**. While Chain-of-Thought (CoT) prompting circumvents the Transformer's architectural limits for such iterative computations, its practical application is plagued by brittleness over long sequences. We propose a new perspective on this failure, identifying an architectural conflict we term **State-Update Interference (SUI)**. We posit that self-attention's inductive bias for global, semantic association can disrupt the localized, state-dependent updates required by procedural algorithms. Paradoxically, CoT may exacerbate this by unrolling the entire computational history, creating an ever-growing set of distractors that are semantically similar but logically irrelevant, thereby corrupting the state-update process. To investigate this hypothesis, we introduce **Sequential State Quarantining (SSQ)**, a diagnostic instrument designed to isolate this failure mode. SSQ periodically forces the model to compress its reasoning trace into a compact state while discarding the preceding context, surgically enforcing the narrow information bottleneck required for procedural logic. On a suite of algorithmic tasks, SSQ yields dramatic performance gains, with accuracy scaling monotonically with the frequency of this intervention. Our findings suggest that a primary bottleneck for procedural reasoning is architectural: a failure of **context management** that is distinct from general limitations of context length or logical capacity. This reframes the problem, suggesting a need for models that can learn to actively manage their long context. Our source code is provided at an anonymous link.

## 1 INTRODUCTION

Sequential arithmetic tasks such as counting and computing running sums are foundational to algorithmic intelligence (Delétang et al., 2023). Their computational structure is defined by a strict, iterative state-transition dynamic: the state at step $t$ depends exclusively on the state at step $t-1$ (Fischer et al., 1968; Ibarra et al., 2002). This requirement for localized, iterative updates creates a fundamental conflict with the core architectural strength of Transformers (**?**)—their inductive bias for global, long-distance association, which is essential for tasks like open-domain question answering but becomes a liability for procedural reasoning (Figure 1).

Transformers implement a fixed-depth computation, a trait that makes them architecturally unsuited for algorithms requiring a number of sequential updates that scales with input length (Delétang et al., 2023; Zhang et al., 2024). Chain-of-Thought (CoT) prompting (Wei et al., 2022) offers an elegant workaround by shifting the locus of computation from the model's latent weights to its textual output space. By externalizing intermediate steps, CoT enables Transformers to simulate the recurrence needed for these otherwise intractable tasks and even grants them the theoretical capacity for Turing-complete computation (Li et al., 2024c).

Chain-of-Thought (CoT) prompting (Wei et al., 2022) cleverly circumvents this limitation by shifting the locus of computation from the model's latent space to its textual output space (Zhang et al., 2024). By externalizing intermediate reasoning steps, CoT allows Transformers to simulate the recurrent computations needed for tasks that would otherwise be architecturally intractable. Theoretical work has even shown that, under idealized conditions, CoT-augmented LLMs possess the capacity to

simulate Turing-complete computations (Li et al., 2024c), suggesting their upper-bound capabilities are immense.

Yet, a stark gap persists between this theoretical potential and empirical reality. On long-sequence arithmetic tasks, LLMs still fail systematically. We posit this failure stems from a core architectural conflict we term **State-Update Interference (SUI)**. The self-attention mechanism, designed to form a fully-connected graph over its context, cannot easily learn to ignore the vast, logically irrelevant history of prior calculations. Instead of focusing computation on the current state update, attention "leaks" to semantically similar past states, forming spurious dependencies that corrupt the delicate arithmetic logic. Paradoxically, standard CoT exacerbates this vulnerability by unrolling the entire computational history into the context, providing an ever-larger set of distractors that actively misdirects computation.

While prior work has identified general failure modes in long contexts, such as "context dilution" or "positional decay" (Liu et al., 2023; Li et al., 2024a; An et al., 2024), our SUI hypothesis proposes a specific and active mechanism that is particularly acute for procedural tasks. SUI complements theories of passive information loss by describing an **active misdirection of computation**, where the model's associative bias forms high-confidence connections to logically irrelevant past states, directly poisoning the state-update process.

To test our hypothesis, we introduce **Sequential State Quarantining (SSQ)**, a **diagnostic instrument** designed to create a near-perfect, surgically-ablated information bottleneck. It is crucial to distinguish the intent of SSQ from performance-oriented heuristics like sliding-window context management. Whereas such methods are efficiency-driven approximations that do not guarantee the preservation of the logical state, SSQ is an experimental intervention. At periodic intervals, we prompt the LLM to *compress* its verbose reasoning trace into a compact, canonical state, *discard* the preceding context, and *resume* computation conditioned only on this quarantined state. The goal is not to propose a practical method, but to create a *controlled* condition that manually enforces the narrow dependency frontier required by iterative algorithms, thereby isolating the effects of SUI.

Our experiments yield compelling results. SSQ dramatically improves accuracy on long arithmetic sequences, with performance scaling monotonically with the frequency of quarantining. These findings provide strong evidence that the dominant bottleneck is **architectural**—a conflict between the model's design and the task's structure—rather than a deficiency in latent logical capacity. This

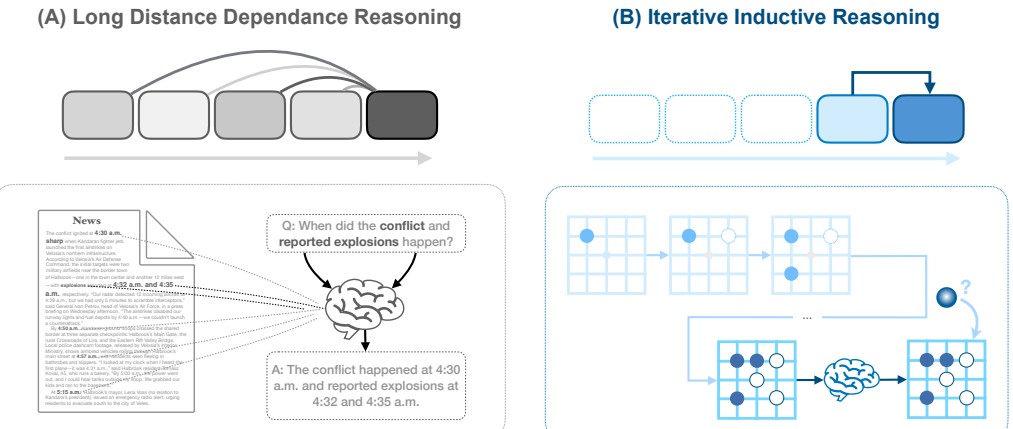

Figure 1: A conceptual distinction between two classes of reasoning tasks. **(A) Long-Distance Dependence Reasoning**, such as open-domain question answering, requires retaining a broad, non-local historical context to synthesize information from multiple, distant points in a sequence. **(B) Iterative Inductive Reasoning**, the focus of this paper, involves tasks with iterative procedural properties where the next state depends *only* on the current state. While models are expected to leverage broad context for tasks in (A), we argue that for tasks in (B), this same architectural bias for global association becomes a liability, causing State-Update Interference (SUI) by attending to logically irrelevant history.

diagnosis points toward new research directions, such as training models with regularization that encourages learned state compression or designing architectures that manage context via disciplined abstraction.

The primary contributions of this paper are therefore:

- We identify and formalize **State-Update Interference (SUI)** as a specific, architecturally-grounded failure mode limiting the effectiveness of CoT on long sequential arithmetic tasks.

- We introduce **Sequential State Quarantining (SSQ)**, a diagnostic instrument designed as a targeted experimental intervention to empirically validate our SUI hypothesis.

- We provide strong evidence that the core limitation is not an inability to perform the underlying logic but rather the architectural bias of self-attention, offering a new perspective to guide the development of more robust procedural reasoning models.

## 2 ARCHITECTURAL LIMITS ON SEQUENTIAL COMPUTATION

The challenge of teaching neural networks to perform algorithmic reasoning is not merely a matter of scale, but one that reveals deep-seated architectural conflicts (Chang & Bisk, 2024). The systematic failures of modern Transformers on these tasks are not accidental but are a direct consequence of an architectural design that is fundamentally misaligned with the nature of sequential, state-dependent computation. This section dissects this misalignment by contrasting the Transformer's design with that of recurrent architectures, thereby establishing the necessary precursors for the **State-Update Interference** phenomenon we diagnose.

### 2.1 THE RECURRENT INDUCTIVE BIAS FOR ALGORITHMIC TASKS

Recurrent Neural Networks (RNNs) and their variants, such as LSTMs, possess a strong inductive bias for sequential processing. Their architecture natively implements the state-transition dynamics $S_{t+1} = f(S_t, x_t)$ through a recurrent update rule:

$$h_t = f_\theta(h_{t-1}, x_t). \tag{1}$$

This structure provides a natural mechanism for maintaining and updating a compact, internal state $h_t$. Early work demonstrated that RNNs could learn to recognize regular languages like $a^n b^n$, which implicitly requires counting (Rodriguez et al., 1999). LSTMs were later shown to handle more complex dynamic counting, such as balancing brackets, by leveraging their gating mechanisms (Suzgun et al., 2019).

Theoretically, this recurrent connection acts as an information bottleneck, forcing the model to compress all relevant history into the state vector $h_{t-1}$. This architectural prior is crucial for learning generalizable algorithms. As models are trained on longer sequences, they can undergo an **implicit representational merger**, where hidden states from functionally equivalent histories converge, effectively learning a compact **deterministic finite automaton (DFA)** within their latent space (Weiss et al., 2018). This allows them to achieve robust generalization far beyond their training data.

### 2.2 THE CONSTANT-DEPTH LIMITATION OF TRANSFORMERS

In stark contrast, Transformers lack an intrinsic recurrent state. As systematically demonstrated by Delétang et al. (2023), Transformers consistently fail at basic counting and arithmetic tasks where RNNs and LSTMs succeed. This failure is not accidental but is a direct consequence of their architecture. A Transformer's computational depth is fixed by its number of layers, $L$, regardless of the input sequence length $N$ (Li et al., 2024b; Zhang et al., 2024). This creates a fundamental mismatch between the model's fixed-depth parallel processing ($\mathcal{D}_{\text{Transformer}} = \mathcal{O}(L) = \mathcal{O}(1)$) and the linear sequential depth required by algorithmic tasks ($\mathcal{D}_{\text{task}} = \Theta(N)$).

This limitation places vanilla Transformers in the complexity class $\text{TC}^0$, rendering them theoretically incapable of solving even basic counting tasks that require unbounded sequential updates (Li et al., 2024b). More critically, the self-attention mechanism endows the model with an **unconstrained,**

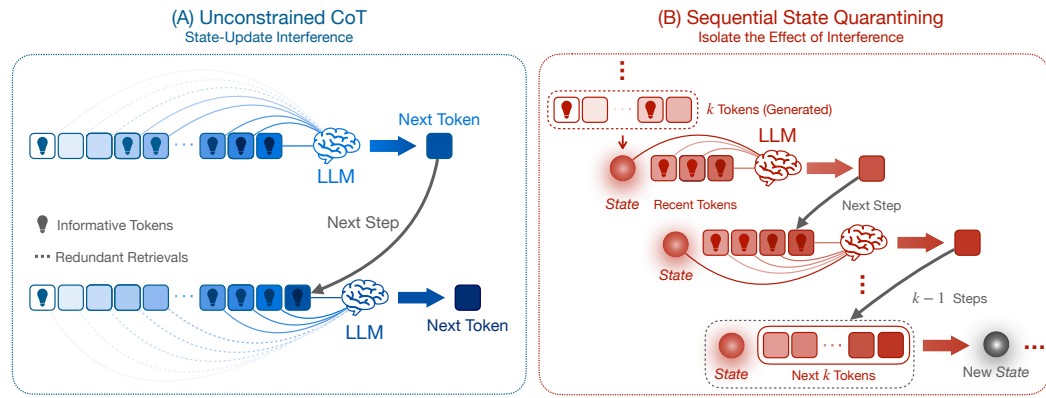

Figure 2: A diagnostic framework for State-Update Interference (SUI). We contrast two settings. **(A) Control (Unconstrained CoT):** In the standard setting, reasoning unfolds as a continuous chain, where spurious self-attention connections (dotted lines) to the full history can corrupt the current state. **(B) Intervention (Sequential State Quarantining):** Our diagnostic method enforces discrete state transitions. A compact state is expanded for the current reasoning step and then re-compressed into a new state, explicitly discarding the intermediate context. This quarantining process ablates historical distractors, allowing us to isolate and measure the performance degradation caused by SUI.

**fully-connected computation graph** at each layer. While this is a powerful feature for capturing non-local dependencies in language, it becomes a liability for procedural tasks. The architecture has no native mechanism to enforce the computational locality of a state update; instead, it has an overwhelming bias toward forming global associations, laying the groundwork for interference.

## 2.3 CHAIN-OF-THOUGHT: SIMULATING RECURRENCE AT THE COST OF INTERFERENCE

Chain-of-Thought (CoT) prompting is an ingenious method to overcome the Transformer's fixed-depth limitation (Wei et al., 2022). It allows the model to *simulate* recurrence by externalizing its computational trace into the context window, effectively trading temporal depth for spatial width. The state update $\mathbf{h}_t = \Phi_\theta(\mathbf{h}_{t-1}, \mathbf{x}_t)$ is approximated through a generate-and-reprocess loop:

$$\mathbf{h}_{t-1} \xrightarrow{\text{Decode}} \boldsymbol{o}_t \xrightarrow{\text{Embed}} \mathbf{h}_t \tag{2}$$

where the latent state $\mathbf{h}_{t-1}$ is decoded into textual thoughts $\boldsymbol{o}_t$, which are appended to the context and re-processed to form the next state. While theoretically Turing-complete under ideal conditions (Li et al., 2024c), this simulation strategy is empirically brittle due to its architectural consequences.

Theoretically, this mechanism is exceptionally powerful. Under ideal assumptions—such as perfect state-to-token fidelity and an unlimited token budget—this externalization loop can simulate unbounded computational depth, making CoT-augmented autoregressive models Turing-complete (Li et al., 2024c). However, the architectural consequences of this simulation strategy make it empirically brittle.

First, by performing a **spatial unrolling** of the entire temporal process, the model is never forced to learn a compressed, abstract state representation. At each step $t$, it generates a new textual state $y^{(t)}$ and appends it to an ever-growing history:

$$h_t = \mathcal{F}_\theta(\text{Emb}(y^{(0)} \oplus y^{(1)} \oplus \cdots \oplus y^{(t-1)})), \tag{3}$$

where $\mathcal{F}_\theta$ is the full Transformer forward pass. Retaining the full history prevents the **representational merger** required to form a robust, generalizable automaton, a process that occurs naturally in recurrent models due to their architectural information bottleneck.

Second, and most critically for our diagnosis, this strategy of simulating temporal depth with spatial length lays the entire computational history bare before the self-attention mechanism. This design is not a neutral trade-off; it directly creates the necessary preconditions for the interference we diagnose in this paper. It transforms the model's capacity for global association from a feature into a fundamental flaw for sequential tasks, setting the stage for systematic failure.

**The Architectural Limits**

There is a fundamental conflict in architectural design. RNNs succeed on sequential tasks by leveraging an **enforced computational management** via their recurrent bottleneck. In contrast, constant-depth Transformers must **simulate** recurrence by spatially unrolling the computational trace via CoT. This simulation exposes the entire reasoning history, creating a structural vulnerability to the very interference our work investigates.

## 3 THE HYPOTHESIS: STATE-UPDATE INTERFERENCE

The strategy of simulating recurrence on a spatial canvas gives rise to a specific and pernicious failure mode. While CoT provides the means to perform sequential computation, the Transformer's core architectural bias systematically corrupts the process. In this section, we formalize our hypothesis of **State-Update Interference (SUI)**, arguing that it is an unavoidable consequence of applying a globally associative architecture to a task that demands locally focused computation.

### 3.1 THE DICHOTOMY OF SEQUENTIAL REASONING

Reasoning tasks processed by Large Language Models (LLMs) can be broadly categorized into two computational paradigms, which place fundamentally different demands on the underlying architecture.

**1. Long-Horizon Associative Reasoning.** This class of tasks, including open-domain question answering and document summarization, requires the model to identify and synthesize information from disparate, non-contiguous segments of a vast context. The computation at any given step may depend on a complex, non-local subset of the entire history. Formally, generating an output token $y_t$ is a function of a sparse set of past hidden states, $y_t \sim p(\cdot|f_\theta(\{\mathbf{h}_i\}_{i \in \mathcal{I}}))$, where the index set $\mathcal{I} \subseteq \{1, \ldots, t-1\}$ can be arbitrarily distributed. The Transformer architecture, with its self-attention mechanism creating a fully-connected graph over the context at each layer, possesses a strong inductive bias for this paradigm. Its strength lies in its ability to draw global associations, making it exceptionally well-suited for these tasks.

**2. Iterative Inductive Reasoning.** This class, the focus of our work, encompasses algorithmic and procedural tasks like counting, parity checking, and running sums. These tasks are characterized by a strict, often Markovian, state-transition structure. The valid state at step $t$, denoted $\mathfrak{s}_t \in \mathcal{S}$ where $\mathcal{S}$ is the state space, depends exclusively on the immediately preceding state $\mathfrak{s}_{t-1}$ and the current input element $x_t$. This defines a recurrent computation:

$$\mathfrak{s}_t = \Phi(\mathfrak{s}_{t-1}, x_t) \tag{4}$$

where $\Phi : \mathcal{S} \times \mathcal{X} \to \mathcal{S}$ is the state update function. For an LLM to succeed, it must learn to approximate this localized computational graph. However, its innate architectural bias for global association becomes a liability. The model must learn to actively *ignore* the vast, logically irrelevant history, a discipline that runs counter to its core design. This architectural mismatch is the primary source of systematic failure on long-horizon inductive tasks.

### 3.2 FORMALIZING STATE-UPDATE INTERFERENCE (SUI) HYPOTHESIS

We posit that the empirical brittleness of CoT on iterative tasks stems from this fundamental conflict. The CoT process approximates the state transition $\mathfrak{s}_t = \Phi(\mathfrak{s}_{t-1}, x_t)$ by unrolling it into a latent-to-text-to-latent cycle. This can be viewed as composing a decoder $\mathcal{D}_\theta : \mathcal{H} \to \mathcal{T}^*$ (mapping latent states to text) and an encoder $\mathcal{E}_\theta : \mathcal{T}^* \to \mathcal{H}$ (re-embedding the text into a latent state). The successful simulation of one step requires the model to faithfully compute:

$$\mathbf{h}_t \approx (\mathcal{E}_\theta \circ \mathcal{D}_\theta)(\mathbf{h}_{t-1}) \tag{5}$$

For this simulation to be robust, the self-attention mechanism must learn to create a **virtual information bottleneck**. That is, when computing $\mathbf{h}_t$, it must isolate its focus, attending almost exclusively to the tokens representing the most recent state $\mathfrak{s}_{t-1}$ and ignoring all prior history. However, the Transformer's associative inductive bias makes this attentional discipline difficult to maintain, leading to **State-Update Interference (SUI)**.

**Attentional Leakage.** The query-key similarity at the heart of self-attention is optimized to find semantic, not procedural, relationships. In tasks like counting, textual representations of adjacent states are often highly similar (e.g., '"The count is 42"' vs. '"The count is 41"'). Let the full context at step $t$ be a sequence of tokens partitioned into disjoint sets $\{\mathcal{C}_k\}_{k=0}^{t-1}$, where each $\mathcal{C}_k$ contains the token indices for the textual representation of state $\mathfrak{s}_k$. When computing the next state, a query vector $\mathbf{q}$ generated from the current context will exhibit high similarity not only with keys $\{\mathbf{k}_i\}_{i \in \mathcal{C}_{t-1}}$ from the correct antecedent state but also with keys from older, logically irrelevant states $\{\mathbf{k}_j\}_{j \in \mathcal{C}_k, k < t-1}$.

This causes the attention distribution to "leak" across the desired computational boundary. Instead of retrieving information solely from the values associated with $\mathcal{C}_{t-1}$, the resulting representation becomes a contaminated mixture. The output of an attention head, $\mathbf{z}$, can be decomposed as:

$$\mathbf{z} = \underbrace{\sum_{i \in \mathcal{C}_{t-1}} \alpha_i \mathbf{v}_i}_{\text{Target State Information}} + \underbrace{\sum_{k=0}^{t-2} \sum_{j \in \mathcal{C}_k} \alpha_j \mathbf{v}_j}_{\text{Interference Term: } \boldsymbol{\epsilon}_{\text{SUI}}} \tag{6}$$

The resulting state is not a clean update but is polluted by the interference term $\boldsymbol{\epsilon}_{\text{SUI}}$, a weighted average of logically invalid prior states. This directly corrupts the fidelity of the simulated recurrence.

**Compounding Distraction via Spatial Unrolling.** The CoT methodology inadvertently creates the ideal conditions for this failure. By spatially unrolling the entire computational history, CoT provides an ever-growing set of distractors. At each step $t$, the number of historical token sets, $|\{\mathcal{C}_k\}_{k=0}^{t-2}|$, grows linearly. This increases both the probability and the potential magnitude of the interference term $\boldsymbol{\epsilon}_{\text{SUI}}$. Paradoxically, the very mechanism that grants the Transformer its theoretical power for sequential computation is also what systematically undermines it in practice. The model is not just failing to attend correctly; it is being architecturally compelled to integrate a growing history of distracting information that poisons the delicate state-update logic.

**Ruling Out Alternative Explanations.** Our SUI hypothesis is distinct from other potential failure modes. It is not merely the **accumulation of serialization errors** (i.e., imperfectly writing a state to text), but a flaw in the computational *process* itself; even with perfect state representation, self-attention would still form spurious connections. Nor is it a problem of **context window limits** or passive information decay. SUI is an *active misdirection of computation* that arises from a qualitative failure of context mismanagement, often occurring long before the context window is exhausted. The act of extending the context via CoT actively exacerbates the problem by providing more distractors, making attentional misdirection increasingly likely.

---

**The Hypothesis**

**State-Update Interference (SUI)** is an architectural failure mode arising from the conflict between a Transformer's associative bias and the demands of localized, sequential logic. When simulating recurrence via CoT, the model fails to maintain a virtual information bottleneck. Its attention leaks to semantically similar but logically irrelevant past states, contaminating the state-update operation with an interference term $\boldsymbol{\epsilon}_{\text{SUI}}$. This is a fundamental failure of attentional discipline, not memory capacity or representational fidelity.

---

## 4 A DIAGNOSTIC FRAMEWORK FOR QUANTIFYING INTERFERENCE

To empirically validate the **State-Update Interference (SUI)** hypothesis, we introduce a diagnostic framework designed to quantify its impact by surgically manipulating the conditions under which it occurs. The core of this framework is an experimental intervention we call **Sequential State Quarantining (SSQ)**, used not as a novel performance-enhancing method, but as a diagnostic probe to test our hypothesis. By systematically ablating the historical context—the very substrate of interference—we can measure the performance gap attributable to this architectural flaw and test whether the model's underlying logical capacity is sound when its biases are constrained.

## 4.1 METHODOLOGY: CONTROL VS. INTERVENTION

Our framework contrasts two conditions to isolate the effect of interference.

**Control (Unconstrained CoT):** The baseline condition uses a standard Chain-of-Thought process. The model's reasoning unfolds in a single, continuous chain, where the context buffer is recursively extended: $\mathcal{C}_k = \mathcal{C}_{k-1} \oplus y_k$. This ever-expanding history maximizes the potential for interference, as self-attention is free to form spurious associations with any past state.

**Intervention (SSQ):** Our intervention, **Sequential State Quarantining**, transforms the reasoning process into a discrete-time state transition system, manually enforcing the information bottleneck that Transformers architecturally lack. The process unfolds in a two-phase cycle: a **State Expansion** phase, where the model generates a reasoning trace conditioned *only* on the previously quarantined state $\mathfrak{s}_{k-1}$ and the current input chunk $X_k$; and a **State Compaction** phase, where this verbose trace is immediately distilled into a new state $\mathfrak{s}_k$. In our experiments, the state compression operator $\sigma_\phi$ is implemented via a simple, fixed-template prompt that instructs the model to summarize the outcome of the preceding trace into a canonical format (e.g., "The current count is now X"). The goal is not to engineer an optimal compression scheme, but to create a reliable information bottleneck for diagnostic purposes. This cycle (formalized in Algorithm 1) surgically severs the model's access to its own distracting history, shielding each computational step from interference.

## 4.2 MEASURING THE INTERFERENCE EFFECT

We quantify the performance cost of SUI by measuring the accuracy gap between our intervention (SSQ) and the baseline (CoT). This interference effect, $\Delta_{\text{SUI}}$, is defined as the average difference in accuracy:

$$\Delta_{\text{SUI}} := \mathbb{E}_{\text{task}} \left[ \text{Accuracy}(\text{SSQ}) - \text{Accuracy}(\text{CoT}) \right]. \tag{7}$$

> **The Diagnosis**
>
> **A large, positive $\Delta_{\text{SUI}}$ would provide strong evidence that State-Update Interference, rather than a general deficit in reasoning, is the dominant bottleneck for Transformers on long-sequence procedural tasks.**

## 4.3 EXPERIMENTAL DESIGN

To empirically dissect the State-Update Interference (SUI) hypothesis, we designed a diagnostic stress test for Transformers. Our methodology uses a "clean room" of synthetic algorithmic tasks to isolate the architectural friction caused by unconstrained historical context, allowing us to directly measure the model's procedural reasoning capabilities when its associative biases are challenged.

We selected three canonical procedural algorithms (Delétang et al., 2023) designed to be maximally susceptible to the hypothesized interference: **COUNT**, which tests the fidelity of iterative arithmetic updates; **PARITY CHECK**, which tests the stable maintenance of a categorical state; and **CYCLE NAVIGATION**, which tests adherence to rule-based state transformations. For each task, an input of length $L$ demands exactly $L$ correct state transitions, making $L$ a direct proxy for the length of the reasoning chain and the cumulative potential for interference.

Our experimental setup creates a controlled opposition between two conditions. The **Control** condition employs standard Chain-of-Thought (CoT), where the model generates a continuous reasoning trace. This method maximally exposes the model to SUI, as the ever-growing history provides a fertile ground for spurious attentional links. In contrast, the **Intervention** applies our **Sequential State Quarantining** (SSQ) procedure. Here, the reasoning process is fractured into discrete updates; after each step, the state is compacted and the intermediate trace is discarded, thereby enforcing a recurrent-like information bottleneck that starves the attention mechanism of historical distractors. We test our hypothesis on two powerful, instruction-tuned LLMs, `Qwen2.5-72B-Instruct` (Team, 2024; Yang et al., 2024a) and `DeepSeek-R1-Distill-70B` (DeepSeek-AI, 2025), to demonstrate that SUI is a general architectural phenomenon.

Table 1: Accuracy comparison between Standard CoT (Control) and Sequential State Quarantining (SSQ, Intervention). The $\Delta_{\text{SUI}}$ row for each model explicitly calculates the performance change, quantifying the impact of State-Update Interference. Positive values (green) indicate that mitigating interference improves performance. SSQ results correspond to $N_s = 2$ decomposition steps.

| Model | Method | Count | | | Parity Check | | | Cycle Navigation | | | Avg. |
|---|---|---|---|---|---|---|---|---|---|---|---|
| | | $L = 50$ | $L = 80$ | $L = 100$ | $L = 50$ | $L = 80$ | $L = 100$ | $L = 50$ | $L = 80$ | $L = 100$ | |
| Qwen2.5-72B-Instruct | Standard CoT | 0.739 | 0.374 | 0.229 | 0.669 | 0.575 | 0.507 | 0.577 | 0.232 | 0.203 | 0.456 |
| | SSQ ($N_s = 2$) | 0.874 | 0.700 | 0.496 | 0.828 | 0.673 | 0.559 | 0.659 | 0.527 | 0.378 | 0.633 |
| | $\Delta_{\text{SUI}}$ | ↑0.135 | ↑0.326 | ↑0.267 | ↑0.159 | ↑0.098 | ↑0.052 | ↑0.082 | ↑0.295 | ↑0.175 | ↑**0.177** |
| DeepSeek-R1-Distill-70B | Standard CoT | 0.615 | 0.124 | 0.062 | 0.644 | 0.301 | 0.231 | 0.377 | 0.062 | 0.051 | 0.274 |
| | SSQ ($N_s = 2$) | 0.845 | 0.608 | 0.352 | 0.745 | 0.570 | 0.479 | 0.551 | 0.317 | 0.172 | 0.515 |
| | $\Delta_{\text{SUI}}$ | ↑0.230 | ↑0.484 | ↑0.290 | ↑0.101 | ↑0.269 | ↑0.248 | ↑0.174 | ↑0.255 | ↑0.121 | ↑**0.241** |

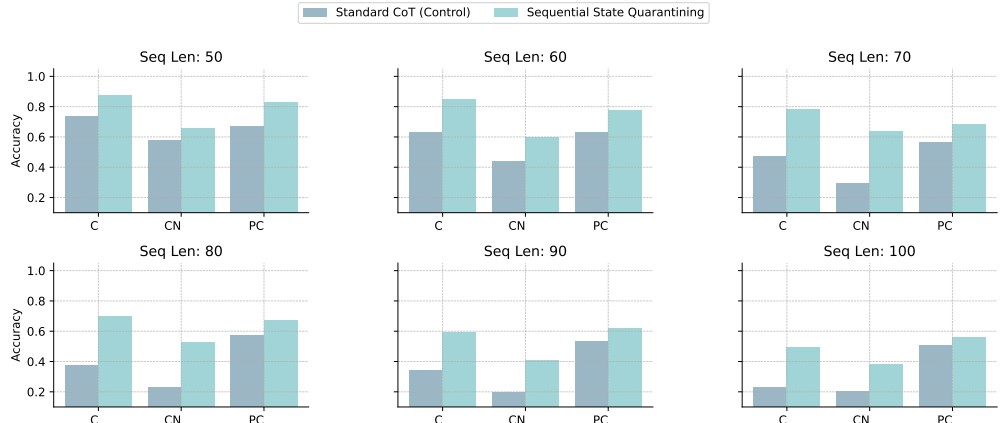

Figure 3: Accuracy as a function of input length $L$ for `Qwen2.5-72B-Instruct`. The performance of standard CoT degrades sharply as length increases, consistent with the SUI hypothesis that a longer history provides more opportunities for interference.

## 5 RESULTS AND DIAGNOSIS VALIDATION

Our experiments yield decisive evidence supporting the SUI diagnosis. The results demonstrate that SUI is not a marginal effect but a dominant performance bottleneck in procedural reasoning. We establish this through two key findings: (1) surgically ablating historical distractors unlocks massive performance gains, and (2) a clear dose-response relationship exists, where more aggressive mitigation of interference leads to monotonically higher accuracy.

### 5.1 SUI IS THE DOMINANT PERFORMANCE BOTTLENECK

As shown in Table 1, enforcing an information bottleneck with SSQ yields consistent improvements over the unconstrained CoT baseline. The performance gap, $\Delta_{\text{SUI}}$, which quantifies the cost of interference, is substantial across all tasks. For `Qwen2.5-72B-Instruct`, the average accuracy gain is **+17.7** points, while for `DeepSeek-R1-Distill-70B`, the recovery is an even more striking **+24.1** points.

This evidence suggests that the primary limitation of these models is not a deficit in their underlying logical "hardware." Rather, their reasoning capabilities are actively suppressed by their architectural design. The catastrophic performance collapse of standard CoT at longer sequence lengths (e.g., DeepSeek's accuracy on COUNT dropping from 61.5% at $L = 50$ to just 6.2% at $L = 100$) aligns perfectly with the SUI hypothesis. As the computational history lengthens, the accumulation of semantically similar distractors overwhelms the attention mechanism, leading to a cascade of errors.

Table 2: **Scalability of SSQ with quarantining frequency** ($N_s$). Accuracy on `Qwen2.5-72B-Instruct` for tasks at lengths $L = 80$ and $L = 100$. The dedicated $\Delta_{\text{SUI}}$ row for each $N_s$ setting shows the absolute accuracy improvement over the Standard CoT baseline. The monotonic increase in these values confirms a strong dose-response relationship between the frequency of interference mitigation and performance.

| Method | Count | | Cycle Navigation | | Parity Check | | Overall Avg. |
|---|---|---|---|---|---|---|---|
| | $L = 80$ | $L = 100$ | $L = 80$ | $L = 100$ | $L = 80$ | $L = 100$ | |
| Standard CoT | 0.374 | 0.229 | 0.232 | 0.203 | 0.575 | 0.507 | 0.353 |
| SSQ ($N_s = 2$) | 0.700 | 0.496 | 0.527 | 0.378 | 0.673 | 0.559 | 0.556 |
| $\Delta_{\text{SUI}}$ | ↑0.326 | ↑0.267 | ↑0.295 | ↑0.175 | ↑0.098 | ↑0.052 | ↑0.203 |
| SSQ ($N_s = 5$) | 0.803 | 0.686 | 0.715 | 0.367 | 0.825 | 0.711 | 0.685 |
| $\Delta_{\text{SUI}}$ | ↑0.429 | ↑0.457 | ↑*0.483* | ↑0.164 | ↑0.250 | ↑0.204 | ↑0.332 |
| SSQ ($N_s = 10$) | **0.994** | **0.994** | **0.937** | **0.864** | **0.988** | **0.921** | **0.950** |
| $\Delta_{\text{SUI}}$ | ↑0.620 | ↑0.765 | ↑0.705 | ↑0.661 | ↑0.413 | ↑0.414 | ↑0.597 |

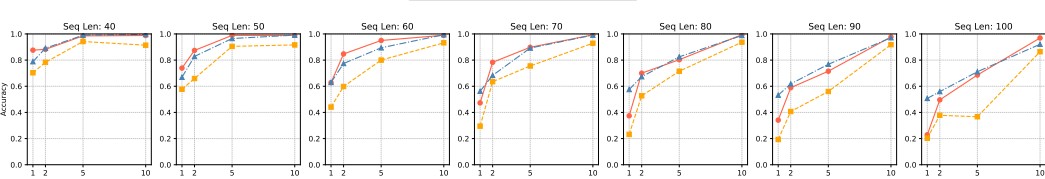

Figure 4: Accuracy as a function of quarantining frequency $N_s$ on `Qwen2.5-72B-Instruct`, averaged across tasks. The strong monotonic improvement demonstrates a clear dose-response relationship, providing robust evidence for the SUI hypothesis.

The large, positive $\Delta_{\text{SUI}}$ values confirm that SSQ is not teaching the model new skills but is simply *un-jamming* a capable reasoning module that was being drowned in attentional noise.

## 5.2 A DOSE-RESPONSE RELATIONSHIP CONFIRMS THE CAUSAL LINK

The most compelling evidence for our diagnosis comes from the clear dose-response relationship between the frequency of interference mitigation and task performance. By varying the quarantining frequency (controlled by the hyperparameter $N_s$, the number of steps per quarantine), we can effectively "titrate" the level of historical interference the model is exposed to.

As predicted, performance scales monotonically with the frequency of intervention. Table 2 and Figure 4 show this effect with striking clarity. For `Qwen2.5-72B-Instruct` on COUNT at length $L = 100$, increasing the quarantining frequency from a low dose ($N_s = 2$) to a high dose ($N_s = 10$) elevates accuracy from a modest 49.6% to a near-perfect 99.4%. This is not merely an improvement; it is a phase transition in capability. The strong, monotonic increase in $\Delta_{\text{SUI}}$ as $N_s$ increases confirms a causal link: more aggressive quarantining of historical context directly translates to higher computational fidelity. This finding solidifies our diagnosis that the primary bottleneck is not an innate inability to perform multi-step reasoning, but rather an architectural predisposition to be distracted by the very computational history that CoT aims to leverage.

## 6 CONCLUSION

In this work, we identified and empirically validated **State-Update Interference (SUI)** as a core failure mode limiting the procedural reasoning capabilities of LLMs. Our diagnostic intervention, Sequential State Quarantining (SSQ), demonstrates that this is not a deficit in logical capacity but an architectural conflict: the Transformer's intrinsic global attention bias corrupts the local, state-dependent computations required by iterative algorithms. This reframing points toward future work beyond prompting heuristics, focusing on architectures with explicit context management or regularization techniques to foster more disciplined and robust sequential reasoning.

# 7 REPRODUCIBILITY STATEMENT

We have taken several steps to ensure the reproducibility of our work. The models used in our experiments, Qwen2.5-72B-Instruct and DeepSeek-R1-Distill-70B, are publicly accessible. Our experiments are conducted on a suite of synthetic algorithmic tasks—COUNT, PARITY CHECK, and CYCLE NAVIGATION—which are based on canonical algorithms from prior work. The methodology for generating these tasks is fully described in Section 4.3, allowing for their exact replication. The implementation details of our proposed diagnostic framework, including the control (Unconstrained CoT) and intervention (Sequential State Quarantining) conditions, are described in Section 4. Key hyperparameters, such as the quarantining frequency ($N_s$), are detailed in Section 5. To facilitate replication, we will release the source code and experiment scripts upon publication. Together, these resources are intended to ensure that our results can be independently verified and our work extended.

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

# A APPENDIX

## A.1 RELATED WORK

**Architectural Limits and In-Context Recurrence.** The fixed-depth, feedforward architecture of Transformers is fundamentally misaligned with the unbounded, iterative nature of many algorithms (Delétang et al., 2023; Zhang et al., 2024). Unlike recurrent models, which naturally scale their computational depth with sequence length, Transformers lack native mechanisms for state persistence or true recursion (Dziri et al., 2024; Valmeekam et al., 2022). Chain-of-Thought (CoT) prompting (Wei et al., 2022) and its derivatives (Yao et al., 2023; Nye et al., 2021; Kojima et al., 2022) have emerged as a powerful paradigm to circumvent this limitation. By externalizing the computational trace into the context window, these methods allow the model to simulate recurrence. Theoretical work has shown that this approach grants Transformers the capacity to simulate Turing machines under idealized conditions (Pérez et al., 2021; Li et al., 2024c), suggesting immense computational potential. However, our work investigates the practical breakdown of this simulation, positing that the very mechanism of in-context unrolling creates the conditions for the attentional failures we term SUI.

**Diagnosing Failures in Long-Context Reasoning.** The performance degradation of LLMs over long sequences is a widely recognized problem. Much prior work attributes this to passive information loss, such as the "lost in the middle" phenomenon where models struggle to retrieve information from the center of their context window (Liu et al., 2023), or a general decay in attentional acuity over distance (Li et al., 2024a; An et al., 2024). These failure modes, often studied in the context of information retrieval or summarization, characterize the problem as a passive decay of signal. Our State-Update Interference (SUI) hypothesis complements these findings by proposing a more **active failure mechanism** that is particularly acute in procedural tasks. We argue that the problem is not just that the model *loses* the correct state, but that it is actively *misdirected* by its own architectural biases to incorporate irrelevant past states into the computation. This distinction is critical: for sequential arithmetic, successful reasoning requires actively *ignoring* historical context, a direct contradiction to the associative capabilities needed for tasks like multi-hop QA (Biran et al., 2024; Yang et al., 2024b; Yoran et al., 2023) where synthesizing distant information is paramount.

**Approaches to Managing Context and State.** A variety of techniques have been developed to improve LLM performance on complex, multi-step tasks. One major line of research focuses on scaling the context window length through architectural modifications like sparse attention (Beltagy et al., 2020; Kitaev et al., 2020; Zaheer et al., 2020) or more efficient key-value caching (Zhang et al., 2023; Fu et al., 2024). These efforts primarily address the *computational cost and memory limits* of long contexts, but they do not necessarily resolve the underlying issue of how attention is allocated within that context. SUI can occur long before the context window is exhausted, suggesting that simply extending the window size may not prevent logical errors. Another line of work seeks to impose structure on the reasoning process through techniques like task decomposition (Zhou et al., 2022; Khot et al., 2022; Drozdov et al., 2022) or by organizing thoughts into trees (Yao et al., 2024; Long, 2023) and graphs (Besta et al., 2024; Sel et al., 2023). While these methods provide valuable scaffolding for complex reasoning, they often treat the LLM's step-by-step execution as a black box. Our work differs by proposing a specific, mechanistic failure *within* that black box, and we use Sequential State Quarantining (SSQ) not as a performance-enhancing heuristic, but as a diagnostic tool to isolate and verify this internal failure mode.

## A.2 EXPERIMENTAL TASKS FOR DIAGNOSING SUI

We evaluate our State-Update Interference (SUI) hypothesis using three procedural reasoning tasks. These tasks, adapted from prior work on algorithmic reasoning (Delétang et al., 2023), are intentionally simple, requiring only basic arithmetic and state tracking. Their simplicity is a key feature of our diagnostic approach; it ensures that model failures are attributable to architectural limitations in handling iterative state updates, rather than a lack of complex problem-solving ability. Each task is designed around a minimal, well-defined state that must be accurately propagated through a sequence of operations, making them ideal for exposing the effects of attentional misdirection. Each task embodies two key properties:

1. **Strict Iterative State Updates:** All tasks follow a Markovian state-transition process, where the state at step $t$ depends exclusively on the state at step $t-1$. This demands that the model maintains a virtual information bottleneck, focusing its computation locally and ignoring the long tail of historical context.

2. **High Inter-State Similarity:** In a standard Chain-of-Thought trace, the textual representations of consecutive states are highly repetitive and semantically similar (e.g., "the current count is 41", "the current count is 42"). This creates a challenging scenario for self-attention, providing a fertile ground for the attentional leakage SUI describes, where queries are likely to form spurious connections with logically obsolete but textually similar past states.

**Character Counting** For this task, the objective is to count the occurrences of the character 'a'. The state $\mathfrak{s}_k$ is a straightforward integer representing the cumulative count at the end of the $k$-th cycle. The prompt provides the model with the count from the previous state, $\mathfrak{s}_{k-1}$, and the current sub-list of characters. The state projection operator, $\sigma_\phi$, then parses the model's output trace $\tau_k$ for the concluding "'Result: ¡number¿"' marker and extracts the updated integer count to serve as the new quarantined state, $\mathfrak{s}_k$.

**Modular Arithmetic (Cycle Navigation)** This task requires the model to track its position within a 5-state cycle. The state $\mathfrak{s}_k$ is an integer representing the agent's position (from 0 to 4) after processing the $k$-th chunk of movements. The model is prompted with its starting position from the prior state, $\mathfrak{s}_{k-1}$, and the list of movements for the current cycle. Similar to the counting task, $\sigma_\phi$ uses a regular expression to extract the final integer position from the "'Result: ¡number¿"' tag in the model's generation, which becomes the next quarantined state.

**Parity Checking** This task introduces a distinction between the state maintained by the SSQ framework and the direct output of the LLM. The goal is to determine if the total count of 'a's is even or odd. The true state tracked by the SSQ protocol, $\mathfrak{s}_k$, remains the cumulative **integer count**. In each cycle, the model receives the integer count from the previous state $\mathfrak{s}_{k-1}$ and is instructed to reason about the final parity, concluding with a **boolean** value ('Result: True' for even, 'Result: False' for odd). This design specifically tests the model's final logical inference step (the parity judgment). The state projection operator $\sigma_\phi$ is responsible for extracting this boolean answer to score correctness, while the SSQ framework updates its internal integer count based on the number of 'a's in the current input chunk to produce $\mathfrak{s}_k$ for the next iteration. This isolates the model's parity logic from the memory burden of tracking the long-range integer state, which is handled by the protocol itself.

## A.3 The SSQ Diagnostic Protocol: Implementation

This appendix details the implementation of our diagnostic protocol, **Sequential State Quarantining (SSQ)**, as specified in Algorithm 1. We elaborate on the core operators and the state-formatting logic used to surgically control the model's context and isolate the effects of State-Update Interference.

### A.3.1 Core Operators

The SSQ protocol is orchestrated by two primary operators that manage the flow of information to and from the language model.

**LLM Generation Operator** ($\mathcal{G}_\theta$) This operator represents a single, **stateless inference call** to the language model, which functions as the black-box reasoning engine under investigation. For our experiments, $\mathcal{G}_\theta$ was an API call to the `gpt-4-turbo-preview` model. The operator takes a formatted prompt string $\mathcal{P}$ as input and returns the model's complete, uninterrupted textual generation $\tau$. To ensure deterministic and reproducible reasoning paths, we set the sampling temperature to 0.0.

**State Projection Operator** ($\sigma_\phi$) This operator is the critical component that **enforces the information bottleneck**. It is a deterministic, non-neural **state projection function** designed to surgically extract a canonical representation of the computational state, $\mathfrak{s}_k$, from the model's verbose reasoning trace, $\tau_k$. For the arithmetic tasks, this function was implemented as a rule-based parser that uses regular expressions to locate a predefined answer marker (e.g., `"{Result: }"`) and extract the

subsequent value. Its deterministic, rule-based nature is essential for the integrity of the diagnostic, as it introduces no new source of model-induced error and guarantees that only the intended state variable is propagated between steps.

**Input Partitioning** The $\text{Partition}(X, N_s)$ function is a straightforward utility that divides the total input sequence $X$ into $N_s$ contiguous, non-overlapping sub-sequences. This partitions the overall task into a series of smaller, state-dependent computations, with each partition $X_k$ being processed in a distinct SSQ cycle.

---

**Algorithm 1** Sequential State Quarantining (SSQ) Protocol

---

**Require:** Initial Prompt $\mathcal{P}_{\text{init}}$, Full Input $X$, Quarantine Frequency $N_s$
**Require:** LLM Generation Operator $\mathcal{G}_\theta$, State Projection Operator $\sigma_\phi$
 1: Initialize quarantined state: $\mathfrak{s}_0 \leftarrow \mathcal{G}_\theta(\mathcal{P}_{\text{init}})$
 2: Partition input: $\{X_1, \ldots, X_{N_s}\} \leftarrow \text{Partition}(X, N_s)$
 3: **for** $k = 1, \ldots, N_s$ **do**
 4:                                ▷ *1. State Expansion (Conditioned Generation)*
 5:     Construct prompt from quarantined state: $\mathcal{P}_k \leftarrow \text{Format}(\mathfrak{s}_{k-1}, X_k)$
 6:     Generate reasoning trace from limited context: $\tau_k \leftarrow \mathcal{G}_\theta(\mathcal{P}_k)$
 7:                           ▷ *2. State Compaction (Surgical Quarantine)*
 8:     Project trace to new state, discarding context: $\mathfrak{s}_k \leftarrow \sigma_\phi(\tau_k)$
 9: **return** Final state/answer $\mathfrak{s}_{N_s}$

---

### A.3.2 PROMPTING AND STATE REPRESENTATION

At each step $k$ of the protocol, the prompt $\mathcal{P}_k$ is dynamically instantiated from a template. This template serves to contextualize the model for the current sub-task, conditioning it exclusively on the most recent quarantined state $\mathfrak{s}_{k-1}$ and the current input chunk $X_k$. The templates used are detailed below, with $\{\{\texttt{variable}\}\}$ denoting placeholders.

**Task 1: Character Counting**

```
Count the number appearances of 'a's in the list below,
starting with a count of '{{count}}'. Think step by step.
Conclude your final answer with: {Result: } followed by the
counted number. For example, if the input list is
['a', 'b', 'a', 'a']', the final output should be concluded
with {Result: 3}.

Start count: {{count}}.
List: {{list}}
```

The $\{\{\texttt{count}\}\}$ placeholder is populated by the quarantined state $\mathfrak{s}_{k-1}$, and $\{\{\texttt{list}\}\}$ is populated by the input chunk $X_k$.

**Task 2: Modular Arithmetic**

```
Given a list of movements on a cycle of length 5, start at
position '{{position}}' and compute the end position. The
movements are STAY, INCREASE, DECREASE and are represented
as {0, 1, 2}.
Please determine the agent's final position after executing
all movements in the list. Think step by step.
Conclude your final answer with: {Result: } followed by the
final position. For example, if the input list is
['0', '1', '2', '1']', the final output should be concluded
with {Result: 1}.
```

```
Start position: {{position}}
List: {{list}}
```

Here, {{position}} is replaced by the state $\mathfrak{s}_{k-1}$.

**Task 3: Parity Checking**

```
Determine whether the number of occurrences of letter '{{letter}}'s
in the list below is even, starting with a count of '{{count}}'.
Think step by step.
Conclude your final answer with: {Result: True} if the count is
even, {Result: False} if the count is odd. For example, if the
input list is ['a', 'b', 'a', 'a']', the final output should be
concluded with {Result: True}.

Start count: {{count}}
List: {{list}}
```

This task requires careful state management by the framework. The true state tracked by the SSQ protocol is the cumulative numeric count. However, the LLM is prompted to reason about parity and output a boolean. The state projection operator $\sigma_\phi$ extracts this boolean judgment. The framework then updates its internal, ground-truth count based on this judgment, ensuring the state passed to the next iteration ($\mathfrak{s}_k$) is the correct integer count. This design isolates the LLM's logical inference from the protocol's state-tracking machinery.

### A.3.3 EXECUTION TRACE: CHARACTER COUNTING EXAMPLE

We trace one full cycle of the SSQ protocol for the character counting task to illustrate the information flow.

1. **Initialization:** The protocol begins with an initial state $\mathfrak{s}_0 = 0$. The full input list is partitioned; assume the first partition is $X_1 = [\,'a',\ 'c',\ 'a',\ 'a'\,]$.

2. **Cycle 1: State Expansion & Generation:** The prompt $\mathcal{P}_1$ is constructed by populating the template with $\mathfrak{s}_0$ and $X_1$:

    ```
    Start count: 0.
    List: ['a', 'c', 'a', 'a']
    ```

    The operator $\mathcal{G}_\theta$ is invoked with $\mathcal{P}_1$, producing the reasoning trace $\tau_1$:

    ```
    Okay, I will count the 'a's.
    - Start with count = 0.
    - First element is 'a', count is now 1.
    - Second element is 'c', count is still 1.
    - Third element is 'a', count is now 2.
    - Fourth element is 'a', count is now 3.
    The final count for this list is 3.
    {Result: 3}
    ```

3. **Cycle 1: State Compaction & Quarantine:** The state projection operator $\sigma_\phi$ parses the trace $\tau_1$. It matches the pattern {Result: 3} and extracts the integer 3. This value becomes the new quarantined state, $\mathfrak{s}_1 = 3$. The context from this cycle, including $\mathcal{P}_1$ and $\tau_1$, is now discarded entirely.

4. **Cycle 2: Next Iteration:** The protocol proceeds to the next input partition, $X_2 = [\,'b',\ 'a',\ 'd',\ 'a'\,]$. A new prompt, $\mathcal{P}_2$, is constructed using the newly quarantined state $\mathfrak{s}_1 = 3$:

    ```
    Start count: 3.
    List: ['b', 'a', 'd', 'a']
    ```

This cycle of expansion and compaction continues until all partitions are processed. The final state, $\mathfrak{s}_{N_s}$, is the result.

## A.4 THE USE OF LARGE LANGUAGE MODELS (LLMS)

Large Language Models (LLMs) served as assistive tools for improving the clarity and grammar of our academic prose. Specifically, we leveraged GPT-4o for drafting and refining sections such as the introduction and method. The authors retain full responsibility for all scientific content, including the conception of the research questions, methodological contributions, and the validation of experimental results.

