# OpenReview forum: "Why Do LLMs Fail at Arithmetic Logic? A Diagnosis of Limits on Sequential Computation"
_ICLR.cc/2026/Conference — ICLR 2026 Conference Withdrawn Submission_

### Official Review · Reviewer_Ko9e · 2025-10-22

**Soundness:** 2
**Presentation:** 2
**Contribution:** 2
**Rating:** 2
**Confidence:** 4

**Summary:**

The paper studys the problem that Transformer-based (large) language models fail at sequential arithmetic tasks even with very simple logic (e.g., counting, parity-check).
The authors propose a perspective, dubbed as "State-Update Interference" (SUI), to view these failures: the entire generation process creates the ever-growing set of distractors (states) with similar semantics but irrelevant logic, which can corrupt the state-update process.
They further investigate the SUI hypothesis with a diagnostic tool, Sequential State Quarantining (SSQ), which forces the model to compress the reasoning trace (preceding context) into a single state and generate the follow-up reasoning trace only conditioned on this compacted state (information bottleneck).
The experiments show that SSQ yields obvious performance gains on several simple algorithmic tasks.

**Strengths:**

1. The topic studied in the paper is important, demonstrating that current transformer-based LLMs still can not learn robust algorithmic solution even for very simple sequential arithmetic tasks like counting. The paper writing is overall clear and concise.

2. The proposed SSQ could effectively improve models' performance on tasks (Count, Parity Check and Cycle Navigation), verifying the SUI hypothesis proposed in the paper. The reviewer also thinks the SUI perspective to view LLMs' failure on these sequential tasks to some extent deepens the community's understanding towards building more robust language models.

**Weaknesses:**

(1) Is this SUI the most dominant factor that accounts for LLMs' reasoning failures on these sequential tasks? Many reasoning tasks contains both long-horizon associative reasoning and iterative inductive reasoning. Even the tasks studied in this paper (e.g. count) require the model to precisely retrieve elements from the long input list. The work seems to claim that most failures stem from the state update (e.g., count = 3 => count = 4) errors. What are the ratios of such kind of failures?

(2) Though I know that SSQ serves as a diagnostic probe for the SUI hypothesis, I do not think reasoning traces for most general tasks could be "compacted" in to a single 'state', which limits the applicablity of the SSQ proposed in the paper.

(3) The research was not conducted in sufficient depth. Existing works (e.g., LightThinker [1], Gist Tokens [2]) have already explored methods of compressing reasoning traces (though their aims are not entirely identical with this paper). The reviewer believes current version of the paper still need to explore more comprehensively to this problem (e.g., what is the root cause of SUI? how to design applicable methods to mitigate these issues?) to contribute more meaningfully to the community.

(4) Some typos. For instances, line 40 (Transformers (?)) missing citation; line 272, incorrect quotation marks.

[1] LightThinker: Thinking Step-by-Step Compression. https://arxiv.org/abs/2502.15589, EMNLP 2025.

[2] Learning to Compress Prompts with Gist Tokens. https://arxiv.org/abs/2304.08467, NeurIPS 2023.

**Questions:**

Is this SUI the most dominant factor that accounts for LLMs' reasoning failures on these sequential tasks?

---

### Official Review · Reviewer_dp1q · 2025-10-28

**Soundness:** 3
**Presentation:** 1
**Contribution:** 3
**Rating:** 4
**Confidence:** 3

**Summary:**

The authors hypothesize that the reason for why transformers fail at sequential tasks is because the long context is counterproductive in these tasks by distracting them with irrelevant information. To investigate this issue, the authors propose SSQ, a testing suite to isolate previous long context from the information processing of the current state. Empirical evidence suggests that indeed LLMs' performance is improved with SSQ.

**Strengths:**

1. The research question about state tracking is fundamental to understanding LLM capabilities. It is also relevant to how LLMs handle memory and comparing transformer architectures to other neural networks such as RNNs.

2. The experiment shows a careful design by leveraging three tasks about state tracking. The empirical results showcase that the authors' hypothesis is valid and the analysis provides relevant insights.

**Weaknesses:**

1. There are repeated sentences/missing citations in the paper (e.g., line 40, 45-53). The authors are encouraged to carefully proofread their paper for better readability.

2. Are the problems investigated in the paper only about tracking one state? What happens if we track multiple states? In general, I wonder how is generalizability of the authors' observations/proposed mechanisms in the context of general tasks.

3. The paper mentions RNN as a success in managing sequential tasks and systematic failure of transformers in handling sequential tasks. It is not immediately intuitive to me as for why the experiment is done on LLMs, which are prone to data contamination, instead of vanilla transformers. Is it possible to compare transformers with RNNs in experiment or is there a compelling reason that the current study should focus on LLMs?

4. It is not entirely surprising that irrelevant context can distract LLMs. The authors are encouraged to discuss more about how their insight is novel compared to existing works. See [1, inter alia].

[1] Shi, F., Chen, X., Misra, K., Scales, N., Dohan, D., Chi, E. H., ... & Zhou, D. (2023, July). Large language models can be easily distracted by irrelevant context. In International Conference on Machine Learning (pp. 31210-31227). PMLR.

**Questions:**

The authors are encouraged to address the questions/concerns in the weakness section.

---

### Official Review · Reviewer_UnvM · 2025-11-03

**Soundness:** 3
**Presentation:** 3
**Contribution:** 3
**Rating:** 4
**Confidence:** 4

**Summary:**

This page investigates why LLMs, despite their strong reasoning capabilities, struggle with simple sequential arithmetic tasks such as counting. The authors identify a core architectural bottleneck they call State-Update Interference (SUI), a conflict between the Transformer’s global self-attention mechanism and the localized, iterative updates required for procedural logic. They propose a diagnostic framework, Sequential State Quarantining (SSQ), which periodically forces the model to compress its reasoning trace into a compact state and discard prior context, effectively isolating each computational step. Experiments on tasks like counting, parity checking, and cycle navigation show that SSQ dramatically improves accuracy, confirming that the primary failure arises not from insufficient logical capacity but from architectural context mismanagement. The study reframes LLM reasoning limitations as a design issue of attention and context control, suggesting that future models should incorporate explicit mechanisms for state compression or context discipline to support robust procedural reasoning

**Strengths:**

- This work rigorously connects the architectural properties of Transformers (fixed-depth, global attention) to limitations in simulating recurrent, state-dependent computation.
- The SSQ framework is a controlled intervention that isolates architectural interference effects rather than serving as a mere performance hack.
- Results across tasks (counting, parity check, cycle navigation) show consistent accuracy improvements.
- Figures are clear.

**Weaknesses:**

- This paper focuses on   simple synthetic benchmarks, failing to capture more realistic reasoning settings.
- Only two models (Qwen2.5-72B and DeepSeek-R1-70B) are evaluated.
- SSQ uses hand-crafted prompts, deterministic parsing rules that is limited in uncontrolled and real-world test cases.

**Questions:**

- Does the finding still hold on more realistic reasoning tasks, not just simple synthetic ones? Can you show results for this?
- Would the results be the same if tested on more or different models?
- Can the SSQ method work in real-world settings without hand-crafted prompts and strict parsing rules?

---

### Note · Authors · 2026-01-02

**Comment:**

We thank the reviewers for their constructive feedback in identifying these limitations, which we will address in the revised version.

**Withdrawal Confirmation:**

I have read and agree with the venue's withdrawal policy on behalf of myself and my co-authors.